# Putting Policy into Practice: How Three Cancer Services Perform against Indigenous Health and Cancer Frameworks

**DOI:** 10.3390/ijerph19020633

**Published:** 2022-01-06

**Authors:** Emma V. Taylor, Marilyn Lyford, Lorraine Parsons, Michele Holloway, Karla Gough, Sabe Sabesan, Sandra C. Thompson

**Affiliations:** 1Western Australian Centre for Rural Health, The University of Western Australia, Geraldton, WA 6530, Australia; marilyn.lyford@uwa.edu.au (M.L.); lorrainemp47@gmail.com (L.P.); michele.holloway@uwa.edu.au (M.H.); sandra.thompson@uwa.edu.au (S.C.T.); 2Department of Health Services Research, Peter MacCallum Cancer Centre, Melbourne, VIC 3000, Australia; karla.gough@petermac.org; 3Department of Nursing, Faculty of Medicine, Dentistry and Health Sciences, University of Melbourne, Parkville, VIC 3010, Australia; 4Department of Medical Oncology, Townsville Cancer Centre, Townsville Hospital and Health Service, Townsville, QLD 4814, Australia; sabe.sabesan@health.qld.gov.au

**Keywords:** Aboriginal and Torres Strait Islander, Indigenous Australians, cancer services, cancer care, cancer control, cultural safety, health systems, framework

## Abstract

Improving cancer outcomes for Indigenous people by providing culturally safe, patient-centred care is a critical challenge for health services worldwide. This article explores how three Australian cancer services perform when compared to two national best practice guidelines: the National Aboriginal and Torres Strait Islander Cancer Framework (Cancer Framework) and the National Safety and Quality Health Service (NSQHS) User Guide for Aboriginal and Torres Strait Islander Health (User Guide). The services were identified through a nationwide project undertaken to identify cancer services providing treatment to Indigenous cancer patients. A small number of services which were identified as particularly focused on providing culturally safe cancer care participated in case studies. Interviews were conducted with 35 hospital staff (Indigenous and non-Indigenous) and 8 Indigenous people affected by cancer from the three services. The interviews were analysed and scored using a traffic light system according to the seven priorities of the Cancer Framework and the six actions of the NSQHS User Guide. While two services performed well against the User Guide, all three struggled with the upstream elements of the Cancer Framework, suggesting that the treatment-focused Optimal Care Pathway for Aboriginal and Torres Strait Islander People with Cancer (Cancer Pathway) may be a more appropriate framework for tertiary services. This article highlights the importance of a whole-of-organisation approach when addressing and embedding the six actions of the User Guide. Health services which have successfully implemented the User Guide are in a stronger position to implement the Cancer Framework and Cancer Pathway.

## 1. Introduction

Aboriginal and Torres Strait Islanders are Australia’s First Peoples, with diverse cultures and languages and a history spanning over 50,000 years. (Hereafter the term “Indigenous Australians” is respectfully used.) Indigenous Australians have a holistic approach to health, which incorporates connection to land, culture, community and identity. However, since European colonisation of Australia and the subsequent disruption of traditional lifestyles, chronic diseases such as cancer have emerged, and cancer is now one of the leading causes of death for Indigenous Australians [1]. Despite overall improvements in cancer outcomes in Australia, cancer incidence and mortality are higher for Indigenous Australians, and the gap in cancer mortality rates is widening [2,3,4].

We know that many factors contribute to poorer cancer outcomes for Indigenous people, including higher levels of modifiable risk factors relevant to cancer, lower participation rates in national cancer screening programs, later stage at diagnosis, the presence of comorbidities, lower uptake and completion of cancer treatment and institutional racism in the health system [2,5,6,7,8]. Furthermore, Indigenous people face a range of challenges to engaging with the health system, including fear or mistrust of mainstream health services, lack of respect or cultural understanding shown by clinicians, experiences of racism and logistical difficulties in accessing treatment services [9,10,11,12]. Therefore, cancer service providers are vital to improving outcomes for Indigenous people with cancer by improving accessibility to cancer services and supporting Indigenous Australians through treatment. This includes providing care that is culturally safe and person-centred and which meets their physical, psychological, social, cultural and spiritual health care needs [13,14]. Attention must focus on how cancer services are delivered to Indigenous patients and on ways health services can better meet the needs of these patients.

Recent Australian government publications including the *National Aboriginal and Torres Strait Islander Cancer Framework* [15] and *The Optimal Care Pathway for Aboriginal and Torres Strait Islander People with Cancer* [16] outline a number of areas to consider when providing quality cancer care to Indigenous people. The *National Aboriginal and Torres Strait Islander Cancer Framework* (Cancer Framework), encompasses the full continuum of cancer control and aims to improve Indigenous cancer outcomes by targeting structural inequities and ensuring that Indigenous people have access to and receive good quality cancer care [15]. It sets out seven priority action areas ranging from community education and prevention to diagnosis and treatment, as well as encompassing support for families and improving data systems and research. The Cancer Framework aims to guide organisations in the development of their own action plans to suit local needs and includes a focus on culturally respectful, evidence-based, multidisciplinary patient-centred care. The *Optimal Care Pathway for Aboriginal and Torres Strait Islander People with Cancer* (Cancer Pathway) provides guidance to health services and health professionals on best practice care for Indigenous people with cancer throughout the patient journey [16]. It outlines seven critical steps in the patient journey, starting with early detection and continuing through to diagnosis, treatment and end-of-life care. However, the Cancer Pathway focuses on the aspects of the cancer care pathway that should be responsive to the needs of Indigenous people with cancer.

While not cancer-specific, in 2017 the Australian Commission on Safety and Quality in Health Care published the second edition of the *National Safety and Quality Health Service Standards* (NSQHS) [17], which included six actions that were designed to specifically meet the needs of Indigenous Australians. The six actions cover areas that Indigenous Australians and health service representatives believe can have the biggest impact on reducing structural inequities and improving the quality of care and health outcomes for Indigenous people. By embedding these actions in the NSQHS Standards, all acute health services must now demonstrate that the actions are being addressed to pass their assessments. The *NSQHS User Guide for Aboriginal and Torres Strait Islander Health* (User Guide) [18] was published to provide health services with practical strategies on how to implement the six Indigenous-specific actions.

The aim of this paper was to explore how three services, which had previously been identified through a comprehensive process as “high performing” in the care they provide to their Indigenous patients and family, measured up when compared to two national best practice guidelines: the Cancer Framework and the User Guide. We wished to identify the areas where the three services are doing well and highlight what activities they are performing in those areas, as this may benefit other services wishing to improve their performance on one or both guidelines. However, we also wished to explore whether there are sections in the guidelines where even high performing services struggle and to identify where additional support or attention may be required.

## 2. Methods

### 2.1. Study Design, Service Selection and Characteristics

This study forms part of a national investigation to identify and describe cancer services providing treatment to Indigenous cancer patients in Australia. Public cancer treatment centres across Australia were surveyed to identify the type of cancer services provided, determine their Indigenous patient numbers and explore policy and implementation of Indigenous-specific initiatives [19]. Surveys were completed by 58 of the 125 public cancer treatment centres contacted. Based on findings from the survey, follow-up interviews were conducted with a subset of service providers to explore current practice and programs aimed at improving cancer care for Indigenous Australians [20]. Finally, centres which reported promising practices were identified, with four services agreeing to participate in more detailed study around their specific practice and innovation. Two of the services have been previously reported, with findings on Indigenous workforce policies and experiences of Indigenous cancer patients and their families [21,22]. One service is still undergoing data collection and analysis and therefore was not able to be included. Therefore, three services were considered appropriate for analysis using the Framework and User Guide. 

Intrinsic case study methodology was used to examine the selected services. With intrinsic case study, the cases are preselected due to their specific features, which make them of particular interest [23]. There is no expectation that the services be “representative of other cases”; rather, their value lies in their “uniqueness” [24].

The three public services represent some of the diversity that exists between health services in Australia. Two services are in capital cities; however, Service A is located within a privately operated 800-bed tertiary hospital which employs 6700 staff, while Service B is a public hospital dedicated to cancer treatment, research and education, with nearly 100 beds, and employs more than 3200 staff. Service C is in a state-run 800-bed public hospital which employs 6400 staff and is located in a large regional centre over 1000 kilometres from the next tertiary hospital. 

### 2.2. Cultural and Ethical Considerations

This study received ethics approval from the Western Australian Aboriginal Health Ethics Committee (WAAHEC) (approval number 483), the Human Research Ethics Committees of University of Western Australia (RA/4/1/6286), St Vincent’s Hospital Melbourne Human Research Ethics Committee (approval number HREC/16/SVHM/94) and each participating site. The research project was conducted under the auspices of the Indigenous-led Centre for Research Excellence (CRE), Discovering Indigenous Strategies to improve Cancer Outcomes Via Engagement, Research Translation and Training Centre of Research Excellence (DISCOVER-TT). We adhered to the National Health and Medical Research Council (NHMRC) Guidelines for Ethical Conduct in Aboriginal and Torres Strait Islander Health Research [25]. An Indigenous Advisory/Reference Group advised the study, and three Indigenous researchers were members of the study team.

### 2.3. Participant Recruitment and Profile

Hospital staff (Indigenous and non-Indigenous) were eligible to participate if they cared for or supported Indigenous cancer patients or if they held a leadership role in the care of Indigenous patients at one of the three sites in the case study. Indigenous adults affected by cancer (including Indigenous people diagnosed with cancer and family members) were eligible if they had experienced or observed cancer care at one of the three participating sites. 

Recruitment was purposive, with staff, Indigenous cancer patients and family members identified and recruited in-person by health service staff within each participating service. The participants were generally selected in conjunction with the site investigator. All participants were initially approached and informed about the research by the site investigator. If permission was granted, these individuals were then contacted by a member of the research team and given detailed study information. All participants gave written or oral consent prior to data collection. Interview participation was voluntary and participants could stop the interview at any time. Indigenous people affected by cancer were encouraged to have a family member present during the interview if they wanted.

In-depth interviews were conducted with relevant hospital staff (n = 35). Ten participants were employed in Service A, eleven participants worked in Service B, and fourteen participants worked at Service C. Over three-quarters of staff participants were women (n = 27; 77%), and almost a quarter (n = 8) identified as Aboriginal or Torres Strait Islander. A diverse set of professions participated, including Indigenous Liaison Officers (ILOs), social workers, cancer coordinators, registered nurses, oncologists, managers, executives and administration staff. 

Six in-depth interviews were conducted with Indigenous patients with cancer (n = 5; four male) and affected family members (n = 3; all female). Three patients and one family member spoke about their experiences at Service A, two patients and two family members described their experiences with Service C and no patients were available to be interviewed at Service B.

### 2.4. Data Collection

Semistructured interviews were conducted between September 2017 and December 2018 with Indigenous people affected by cancer (n = 8) and hospital staff (n = 35) from the three services, and they ranged from 30 to 60 min in length. The research team created two interview guides (one for hospital staff and one for Indigenous people affected by cancer) which guided the direction of the interviews; however, additional questions were occasionally added in response to individual roles and answers to questions. Interviews were conducted with the use of the appropriate interview guide, audio taped with consent and transcribed verbatim. On two occasions, two staff members were interviewed together, and two patients chose to have a family member present during their interview. Interviews took place at a convenient time and location for participants, usually at the health service. Interviews were conducted by one of three authors (ML, LP, SCT) or by a local research assistant. All interviews with Indigenous people affected by cancer were conducted in person by an Indigenous researcher. Two interviewers were Aboriginal women with clinical backgrounds in cancer, one of whom is an experienced researcher. The two non-Indigenous interviewers both have over twenty years’ experience with collaborative research into improving Indigenous health outcomes.

Indigenous people affected by cancer were asked about their experiences at the cancer centre, including issues experienced, and any suggestions for changes or improvements. Hospital staff were asked a range of questions regarding existing policies and programs to improve engagement with Indigenous people, with particular emphasis on cancer patients, including questions on primary prevention and education programs, cultural awareness programs, cultural identifiers and Indigenous staff employment strategies. We did not examine the biomedical or technical aspects of cancer treatment provided at the three services, as that was out of scope.

Information was also gathered from the health service websites, including strategic plans, Reconciliation Action Plans, research reports and information on Executive and Board membership.

### 2.5. Data Analysis

We followed the thematic analysis process described by Green et al. [26] of immersion in the data with rereading, coding (we used a hybrid inductive–deductive coding strategy), categorisation and aggregation of identified themes. Transcripts were deidentified and imported into NVivo 10 software for initial data organisation and analysis. Discussions and additional analysis within the team refined themes and triangulated patient and staff interviews.

The data were then analysed and grouped by three researchers (EVT, MH, ML) according to the seven priorities of the Cancer Framework. This analysis was also informed by the seven steps of the Cancer Pathway (all of which fall within the Cancer Framework), but due to the complete overlap with the Cancer Framework, the Cancer Pathway was not used for categorising the data. Based on our previous findings about the strong and positive influence of the tertiary health services on two of the cancer services studied [21,22], the data were also analysed and grouped using the six actions of the NSQHS User Guide. The overlap and inter-relation between these three national best practice guidelines are represented graphically in Figure 1.

Services were scored using a traffic light system against the components of the Cancer Framework and User Guide. Services scored green (active) if there was evidence collected during the case study that they had been active in that priority or action for a substantial period of time, if they had multiple projects implementing that action or priority or if their activities in that category had broad reach in the community. Services scored amber (emerging) if they were beginning to undertake projects in that area or if they had one project with limited penetration into the community. Services scored red (not a focus) if they had not conducted any projects in that area, if the only activities in the area were informal or ad hoc or if the area was described as a “weakness” by staff at the service.

## 3. Results

Although all three services were initially identified as high performing in their care for Indigenous cancer patients and their families, the three services performed very differently when the data were analysed through the dual lenses of the Cancer Framework and the User Guide. The data collected from the three services during the case studies were mapped to the seven Cancer Framework priorities and six User Guide actions. Services were scored using a traffic light system (green = active; amber = emerging; red = not a focus) against the components of the Cancer Framework and User Guide. The performance and rating of each service are summarised in Table 1 and Table 2, with elaboration and supporting evidence to follow. 

It is important to note that we only considered “high performance” and “best practice care” from a social and supportive care perspective; we are not able to comment on the biomedical or technical aspects of cancer treatment delivered by the cancer services. Furthermore, the services may have been running programs or initiatives that were not observed or reported during the case studies, as the services are dynamic with respect to initiatives and programs while their performance was only assessed during a discrete time period. 

### 3.1. National Aboriginal and Torres Strait Islander Cancer Framework

#### 3.1.1. Priority 1: Improve Knowledge, Attitudes and Understanding of Cancer by Individuals, Families, Carers and Community Members (across the Continuum)

Although Service B and Service C both described activities in this area, this was not a strong focus for any of the services studied. Service B had undertaken a project, led by a local Indigenous artist and an Elder, to support the creation of a cultural artefact made by Indigenous women who were cancer survivors. The artefact was available for use by Indigenous patients going through treatment at the service. This project allowed the service to engage key community people to share their lived experiences about cancer. A cancer-specific orientation brochure for Indigenous cancer patients created by one of the ILOs at Service C aimed to make culturally appropriate cancer information more available. This brochure contained “*A lot of pictures. What stuff, what to bring. You know, what radiation is, what chemo is*.” (Participant 24, ICC, Indigenous). However, a staff member at Service C encouraged cancer survivors to “*educate [their] own mob… because there is nobody else doing it*” (Participant 17, ILO, Indigenous). 

**Figure 1 ijerph-19-00633-f001:**
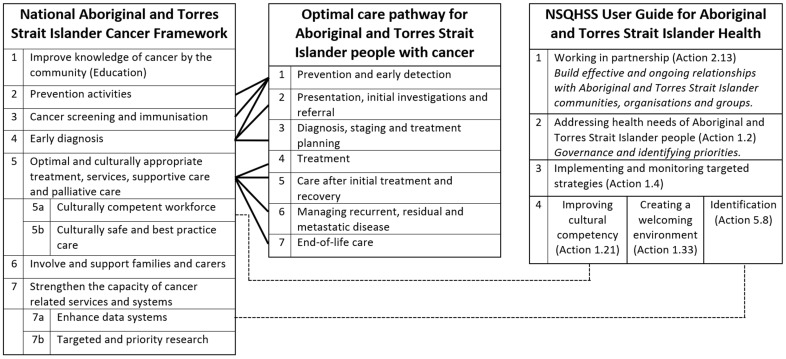
Overlap and inter-relation between the National Aboriginal and Torres Strait Islander Cancer Framework, the Optimal Care Pathway for Aboriginal and Torres Strait Islander People with Cancer and the NSQHSS User Guide for Aboriginal and Torres Strait Islander Health.

**Table 1 ijerph-19-00633-t001:** Cancer service performance scored against the National Aboriginal and Torres Strait Islander Cancer Framework priorities using traffic light rating system.

	Priorities	Service A	Rating	Service B	Rating	Service C	Rating
1	Improve knowledge of cancer by the community (Education)	Staff unsure whether their organisation was involved in education	Not a focus	A cultural artefact made by cancer survivors is available for use by Indigenous patients undergoing treatment. Staff unsure whether their organisation was involved in education	Emerging	Cancer information and orientation brochure created by ILO Ad hoc encouragement of survivors to educate their community Staff unsure how education should be approached in regional and remote areas	Emerging
2	Prevention activities	Not reported in the data	Not a focus	Staff felt their organisation should be involved in prevention, but that it would require a “proper project”	Not a focus	Telehealth antismoking program with remote communities	Active
3	Cancer screening and immunisation	Not reported in the data	Not a focus	Not reported in the data	Not a focus	Long-term breast screening clinic with ILO participation	Active
4	Early diagnosis	Informal network between community ILOs and hospital ILOs	Not a focus	Not reported in the data	Not a focus	Not reported in the data	Not a focus
5	Optimal and culturally appropriate treatment, services, supportive care and palliative care						
5a	Culturally competent workforce	Large number of Indigenous staff Well-supported Indigenous workforce Professional development for Indigenous staff Cultural awareness training for non-Indigenous staff Multidisciplinary care teams	Active	Unable to recruit or retain Indigenous staff Minimal cultural awareness training	Not a focus	Large number of Indigenous staff Well-supported Indigenous workforce Professional development for Indigenous staff Cultural awareness training for non-Indigenous staff Multidisciplinary care teams	Active
5b	Culturally safe and best practice care	Early, “automatic” involvement of ILO with all Indigenous patients ILOs fulfil an informal navigator and care coordinator role Early engagement in end-of-life planning with Indigenous palliative patients Programs to admit patients directly to the oncology ward and provide “one-stop shop” for regional patients	Active	Not reported in the data	Not a focus	Early, “automatic” involvement of ILO with all Indigenous patients ILOs fulfil an informal navigator and care coordinator role Telehealth used to provide chemotherapy treatment and follow-up oncology consultations Early engagement in end-of-life planning with Indigenous palliative patients	Active
6	Involve and support families and carers	Staff asked patients who needed to be kept informed and actively engaged family “even if it is a list a mile long… or advocate [for] a family meeting” (Participant 2, Social Worker, non-Indigenous)	Active	No designated Indigenous family space in the new building	Not a focus	Telehealth regularly used as a means of communicating with families and keeping families involved with patients’ care.	Active
7	Strengthen the capacity of cancer-related services and systems						
7a	Enhance data systems	Processes for accurately identifying and recording Indigenous status Training for staff on how to ask identification questions Data systems enhancements to flag Indigenous status in clinical information systems	Active	Staff not routinely asking whether patients identified Indigenous status not flagged in clinical information systems	Not a focus	Processes for accurately identifying and recording Indigenous status Training for staff on how to ask identification questions Data systems enhancements to flag Indigenous status in clinical information systems	Active
7b	Targeted and priority research	Internal retrospective audits on Indigenous patient identification and care, including one into Indigenous cancer episodes of care	Emerging	Senior management identified a need to improve access to clinical trials for Indigenous patients	Not a focus	Strong published research record on ways to improve Indigenous cancer outcomes, including research on teleoncology and prevention programs	Active

**Table 2 ijerph-19-00633-t002:** Cancer service performance scored against the NSQHSS User Guide for Aboriginal and Torres Strait Islander Health actions using traffic light rating system.

	Actions	Service A	Rating	Service B	Rating	Service C	Rating
2.13	Working in partnership	Memorandums of understanding (MoUs) with local Indigenous health organisations Annual two-day forum to engage with all Indigenous staff members	Active	MoU with state Indigenous health organisation Patient data analysis to understand Indigenous population in catchment	Active	Indigenous representation across their decision-making bodies Indigenous community engagement strategy Annual forum to engage with Indigenous staff and community	Active
1.2	Addressing health needs of Aboriginal and Torres Strait Islander people	Indigenous Australians are identified as a priority patient population in the Strategic Plan Active Stretch Reconciliation Action Plan (RAP) Aboriginal and Torres Strait Islander Employment Plan	Active	Senior management identified that their organisation lacked an Indigenous strategy and plan	Not a focus	Improving Indigenous health outcomes a key component of the Strategic Plan Active Innovate Reconciliation Action Plan (RAP)	Active
1.4	Implementing and monitoring targeted strategies	Progress on the RAP discussed at every Executive meeting Staff aware of RAP and organisation’s policies on improving Indigenous health	Active	Previously a working party had developed some guidelines, but the working party had disbanded and the guidelines were not in use	Not a focus	Progress on the RAP discussed at every Executive meeting Staff aware of RAP and organisation’s policies on improving Indigenous health	Active
1.21	Improving cultural competency	Large number of Indigenous staff Well-supported Indigenous workforce Professional development for Indigenous staff Cultural awareness training for non-Indigenous staff Multidisciplinary care teams	Active	Unable to recruit or retain Indigenous staff Minimal cultural awareness training	Not a focus	Large number of Indigenous staff Well-supported Indigenous workforce Professional development for Indigenous staff Cultural awareness training for non-Indigenous staff Multidisciplinary care teams	Active
1.33	Creating a welcoming environment	Acknowledgement of Country plaques, Indigenous flags, local artwork, posters featuring photos of Indigenous staff members Celebrate significant events on the cultural calendar	Active	No Indigenous artwork or flags displayed in the new building Director described the culture as “very white”	Not a focus	Acknowledgement of Country plaques, Indigenous flags, local artwork, posters featuring photos of Indigenous staff members Celebrate significant events on the cultural calendar Consideration of bush medicine	Active
5.8	Identifying people of Aboriginal and/or Torres Strait Islander origin	Processes for accurately identifying and recording Indigenous status Training for staff on how to ask identification questions Data systems enhancements to flag Indigenous status in clinical information systems	Active	Staff not routinely asking whether patients identified Indigenous status not flagged in clinical information systems	Not a focus	Processes for accurately identifying and recording Indigenous status Training for staff on how to ask identification questions Data systems enhancements to flag Indigenous status in clinical information systems	Active

Some staff were unsure whether their organisation was involved in education. Staff at all three services commented that this was an important issue and felt that their health service should be doing more to educate the community, but were unsure how it should be approached, particularly in regional and remote areas. Meanwhile, multiple patients commented on the need to *“lessen the scariness of cancer”* and stated that *“more information to the Aboriginal and Torres Strait Islander community would be helpful”* (Patient 2).

#### 3.1.2. Priority 2: Focus Prevention Activities to Address Specific Barriers and Enablers to Minimise Cancer Risk for Aboriginal and Torres Strait Islander Peoples

Service C had started focusing on prevention and had implemented an antismoking program using telehealth with Indigenous cancer patients and their families. In this program, the treating medical specialist of a patient diagnosed with a smoking-related health condition delivered antismoking education to the patient’s extended family members through videoconferencing, assisted by local Indigenous health workers and general practitioners (GPs). The program used education materials designed by Indigenous Australians, as well as the patient’s X-rays and CT scans to explain the impact of smoking on the patient’s condition. Referrals to antismoking services and helplines were provided, and any family members who were ready to quit smoking were offered nicotine replacement therapy by the rural-based GP. 

Prevention did not appear to be a focus for either Service A or B; however, staff at Service B commented that prevention was something they had *“started thinking about”* but that they needed a *“proper project”* to get it underway. One senior participant from Service C stated in a 2015 telephone interview that running a prevention program is: 

*…hard but we have to do it… at the moment prevention is left to primary care… cancer centres need to take on tackle prevention. We all see patients and we can do primary and secondary prevention… and really should not need extra funding*.(Participant 13, Oncologist, non-Indigenous)

#### 3.1.3. Priority 3: Increase Access to and Participation in Cancer Screening and Immunisations for the Prevention and Early Detection of Cancers

Service C ran an annual breast screening clinic that travelled out to remote communities. For over twenty years one of the ILOs from the oncology department travelled out to the communities with the clinic staff and walked around the communities to let the women know what was happening and support them through the screening process. It was believed that the visible presence of the ILO raised the profile of the clinics within the community and increased attendance, as well as helping the women who were subsequently referred to the cancer service for treatment because *“when those patients come through [the ILO] already knows some of them”* (Participant 19, RTC, non-Indigenous). 

#### 3.1.4. Priority 4: Ensure Early Diagnosis of Symptomatic Cancers

This did not seem to be an area of focus for any of the three services studied. However, staff at Service A described an informal system whereby local Indigenous primary health care services would often contact the hospital ILO to let them know when one of their patients was coming to the hospital. 

Most patients and family members who were interviewed for the case studies reported “confusing” and “stressful” delays with diagnosis, including delays caused by misdiagnosis and slow communication from a (different) cancer service. In primary health care centres, delays were lengthened when the patient was seen by multiple doctors, or by doctors who were not familiar with the patient, resulting in poor continuity of care and slower follow-up of results. Once diagnosed, participants experienced additional difficulties including receiving misleading information from their local GP about how long they would be required to relocate for tests or treatment and not knowing where to go once they had reached the tertiary health service. 

#### 3.1.5. Priority 5: Ensure Aboriginal and Torres Strait Islander People Affected by Cancer Receive Optimal and Culturally Appropriate Treatment, Services and Supportive and Palliative Care 

##### 5a: Ensure a Skilled and Caring Workforce with Effective Cross-Cultural Communication Skills 

The efforts of Service A and Service C to develop a strong Indigenous health workforce, as well as a skilled and caring workforce, were the focus of a recent article by the authors [21]. This article concluded that a strong Indigenous health workforce can be grown when a health service has specific Indigenous employment policies and measurable targets for employing Indigenous staff, as well as prioritising professional development for Indigenous staff and multidisciplinary care teams which include Indigenous staff, and commits to an inclusive and enabling culture with mandatory cultural awareness training for all staff. 

Services A and C employ a large number of Indigenous staff compared to most other health services in the initial survey of cancer services, including in identified Indigenous roles. At the time of the case study, Service A employed 50 Indigenous staff (0.9% of the total workforce) and Service C employed 241 Indigenous staff (3.74% of the total workforce). Moreover, both services had set measurable targets for the employment of Indigenous staff in their RAP, with both services aiming for population parity, and Service A had an active “Aboriginal and Torres Strait Islander Employment Plan”. 

Both services prioritised professional development for Indigenous staff, with Indigenous staff expected to develop a career plan with their manager and supported to achieve higher qualifications. ILOs were encouraged to specialise and attend training on their specialty. ILOs were respected members of the multidisciplinary team, attending regular clinical meetings and speaking on behalf of Indigenous patients, and were regularly consulted by doctors, nurses and social workers. Indigenous and non-Indigenous staff described the value of joint consultations for Indigenous patients to improve communication and cultural safety. Both services referred to informal mentoring for ILOs, with Service A following a two-way learning model to build capacity of both Indigenous and non-Indigenous staff. Both services provided locally developed cultural awareness training, which was mandatory for all staff, and aimed to increase staff knowledge and understanding of Indigenous peoples’ cultures, histories and achievements. Both services supported the development of the future Indigenous health workforce by providing clinical placements for Indigenous students studying medicine, nursing and allied health. In addition, Service A had implemented a successful Indigenous Cadetship Program and a Graduate Nursing Program.

Staff at Service B described recruiting and retaining of Indigenous health staff as a challenge for their health service. At the time of the case study, the ILO position was vacant and no current or former Indigenous staff members were available for interview. Staff were unsure why their service struggled to retain Indigenous staff, with many commenting that the service had recruited a number of Indigenous health workers, but each staff member had only lasted a few months. One staff member described the situation as *“a roundabout of Aboriginal workers coming through”* (Participant 32, Researcher, non-Indigenous). Staff who were interviewed at Service B expressed concern that Indigenous patients were not receiving important support due to the unfilled vacancy in the ILO role. 

##### 5b: Ensure Aboriginal People Receive Best Practice Care 

The efforts of Services A and C to ensure that Indigenous people receive best practice care, and Indigenous people’s perceptions of that care, have been described previously [22]. This article concluded that positive patient outcomes can be achieved when Indigenous staff are actively involved in the care of Indigenous people, emphasising the importance of ILOs in a navigator role supporting Indigenous people throughout their cancer journey and the provision of more holistic support. 

ILOs were vital and central to the efforts of Service A and Service C to support Indigenous patients throughout their cancer journey. ILOs fulfilled a navigator or care coordinator role, in some circumstances making contact with patients before they arrived at the cancer service. Staff described the involvement of the ILO with any Indigenous patient as “automatic”, and usually occurring on the patient’s first visit, and how the ILO would “follow them through the treatment to provide that support all the way through… and hopefully back home” (Participant 8, Manager, Indigenous). Service A provided a program for people deemed likely to be readmitted to the hospital. This included an Indigenous staff member who could provide support to Indigenous patients on discharge and be a “case manager who can bring them into appointments, take them to the local health service, visit them at home” (Participant 3, Social Worker, non-Indigenous). ILOs helped to put Indigenous patients in contact with supportive care services and provided assistance with organising transport, accommodation and meals for patients and, in some cases, their families. Use of the Supportive Care Needs Assessment Tool for Indigenous Patients (SCNAT-IP tool) was mentioned, but it was not clear whether this was routine practice. One ILO described liaising with general practitioners (GPs) to set up care plans for their Indigenous patients so that they were eligible for transport assistance and other funding. Another ILO emphasised the importance of tailoring the support provided to the specific needs of the individual patient. 

Telehealth was used by Service C to provide chemotherapy treatment, as well as follow-up oncology consultations, closer to home. Registered nurses (RNs) at a number of small rural hospitals and clinics were trained in the administration of chemotherapy, with supervision provided by regional cancer chemotherapy clinical nurses via a telehealth link. Oncology telehealth follow-up consultations for patients in smaller communities, including many Indigenous communities, were supported by local GPs or other suitable health professionals during the consultation and included chemotherapy and community-based palliative care monitoring. This reduced the number of regional visits and enabled remote patients to remain closer to home and their support systems for ongoing treatment. 

Staff at Services A and C recognised the importance of raising awareness and increasing access to palliative care. Staff acknowledged that many people (not just Indigenous) have the misconception that engaging with palliative care services meant that death was imminent, which made conversations about palliative care challenging. The importance of ILO involvement and having conversations early was stressed. Doctors, nurses and social workers helped patients with advanced care directives (ACDs), which gave the patient a formal way of expressing their treatment preferences, beliefs, values and goals at end of life to health care providers and family. Engaging Indigenous palliative patients early in end-of-life planning also assisted timely referrals from palliative care services to community service providers when a home transfer was requested. Indigenous staff described successfully advocating for patients to *“go home to Country to die”* (Participant 17, ILO, Indigenous). 

Staff at all three services were aware that access to the service was a barrier for some Indigenous patients. Both Service A and Service C had implemented strategies to alleviate some of the barriers experienced by patients before starting cancer treatment. Service A allowed patients to be admitted directly to the oncology ward and bypass the emergency department, avoiding the stress associated with emergency departments and reducing wait times. The service also provided a *“one-stop shop”* for regional patients *“so they have their consult the same day as their treatment… so we try and do everything so they don’t have to come back twice”* (Participant 1, Nurse Unit Manager, non-Indigenous). Staff also described providing additional orientation information to Indigenous patients and spending extra time with them to build rapport.

#### 3.1.6. Priority 6: Ensure Families and Carers of Aboriginal and Torres Strait Islander People with Cancer Are Involved, Informed, Supported and Enabled throughout the Cancer Experience 

Staff at Services A and C emphasised the significance of family for Indigenous patients and attempted to accommodate their needs by providing sufficient physical space for the whole family to gather and logistical support to family members such as meal and parking vouchers. Staff at both services discussed the importance of ensuring all family members were kept informed. *“Even if it is a list a mile long that is fine… we can make more than one call if we need to or we can advocate that a family meeting or something might be better to get everyone in to have those frank discussions”* (Participant 2, Social Worker, non-Indigenous). At Service C, where there were often great distances separating patients from families, staff used telehealth as a means of communicating with families and keeping families involved with patients’ care.

#### 3.1.7. Priority 7: Strengthen the Capacity of Cancer-Related Services and Systems to Deliver Good Quality, Integrated Services That Meet the Needs of Aboriginal and Torres Strait Islander People

##### 7a: Enhance Data Systems to Inform Better Outcomes

Both Service A and Service C had made a sustained effort to correctly identify and record Indigenous status. Multiple staff at both services emphasised the importance of asking all patients whether they identified regardless of appearance and identification training was conducted with frontline staff. *“We prompt that question, ‘Are you of Aboriginal and Islander descent?’… It is always prompted, no matter what. Even if you are black or white, they still ask the question.”* (Participant 22, Admin, Indigenous). Staff believed it was an area where their service performed well, a belief that was supported in Service A by past audits. In both cancer services, enhancements to the data systems assisted with workload allocation and ensuring that no patients were missed, with ILOs able to print off a list of all Indigenous patients within the service. In Service A, Indigenous flag stickers were also placed on the front of patients’ files to aid identification. 

Staff at Service B highlighted identification as *“the biggest issue”* for their service because it affected funding and the support they were able to offer patients. One senior staff member observed that funding was based on patient numbers and that if the service could identify every Indigenous patient the service would be eligible for considerably more funding and an increased number of ILO positions. Some staff felt that it was a data or system issue and mentioned that Indigenous status was rarely on the GP referral, whereas others were concerned that patients were choosing not to identify. 

*So I think it is around [Service B] and staff understanding why people don’t want to identify as Aboriginal. It is not just a matter of we are not asking the question. It is like we have to provide a safe environment for our Aboriginal patients to disclose their Aboriginality*.(Participant 25, Director, non-Indigenous)

Staff shared anecdotes of being unaware of patients’ indigeneity and were concerned that it had reduced the cultural safety of their care, or the support services the patients were offered. *“I did not even realise he was Aboriginal… I guess I didn’t ask, but it is not documented anywhere”* (Participant 28, Nurse, non-Indigenous). 

##### 7b: Targeted and Priority Research to Inform Policy, Health Promotion, Service Provision and Clinical Practice 

Although Services A and C both mentioned activities in this area, research was not strongly reported in the data for any of the services studied. It is known that all three services were involved in piloting and evaluating the SCNAT-IP tool, which was used to assess the supportive care needs of Indigenous patients. At the time of the case study, Service C had a strong published research record on ways to improve Indigenous cancer outcomes, including research on teleoncology and prevention programs. Service A described retrospective audits on Indigenous patient identification and care, one into Indigenous cancer episodes of care, and research into the welfare and needs of Indigenous cancer patients. 

### 3.2. Health Standards User Guide

Due to commonalities in the requirements for culturally appropriate care, there is some overlap between the User Guide and the Cancer Framework (see Figure 1). Specifically, User Guide Action 1.21 “Improving cultural competency” has much in common with Cancer Framework Priority 5a “Ensure a skilled and caring workforce with effective cross-cultural communication skill”, and Action 5.8 “Identifying people of Aboriginal and/or Torres Strait Islander origin” has a similar focus to Priority 7a “Enhance data systems to inform better outcomes”. Where the main points of an Action have already been described in detail in the Cancer Framework section, only a summary is provided with reference to the relevant Cancer Framework section. To avoid repetition, we have also combined our reporting of service performance against Actions 1.2 “Addressing health needs of Aboriginal and Torres Strait Islander people” and 1.4 “Implementing and monitoring targeted strategies”; Action 1.2 deals with the setting of priorities and development of strategic plans, whereas Action 1.4 covers the implementation and monitoring of those strategies. 

#### 3.2.1. Action 2.13: Working in Partnership

All three services were making efforts to build effective and ongoing relationships with Indigenous communities, organisations and groups that represented or serviced their catchment populations, albeit in different ways. Service C had published an engagement strategy “Engaging with our Aboriginal and Torres Strait Islander Consumers and Community” and had strong Indigenous representation on their decision-making bodies, whereas Services A and B were focused on developing relationships with Indigenous organisations and stakeholders. 

Only Service C had Indigenous representation across their decision-making bodies, with an Executive Director of Aboriginal and Torres Strait Islander Health, a Board member who is a member of the Indigenous community and the requirement that a member of the Clinical Council must be Indigenous. In addition, the board was advised by an Aboriginal and Torres Strait Islander Health Leadership Advisory Council (ATSIHLAC) comprising senior Indigenous health service staff and an Aboriginal and Torres Strait Islander Community Advisory Council (ATSICAC), a forum of community representatives. Staff saw this level of representation as a point of difference for their health service and something to be proud of. 

Both Services A and B had signed a memorandum of understanding (MoU) with local or state Indigenous health organisations to formalise their partnership and as a sign of *“genuine engagement*”. Social Workers at Service A talked about how they prioritised regularly taking new staff to visit local Indigenous organisations to build relationships between the health services and help staff make personal connections. Service C’s engagement strategy listed several Indigenous agencies with whom they had informal partnerships. 

Services A and C held annual forums. Service C held an annual “Close the Gap Day” forum, attended by staff and community members. The forum had a program of invited speakers, including local Indigenous service providers. Service A held an annual two-day forum to engage with all Indigenous staff members (including support staff). The forum was held offsite, and the first day of the forum was attended by members of the Executive team who talked with and sought feedback from staff. 

*The organisation brings all Aboriginal staff, so not just ALOs or managers, it’s nursing staff, it’s cleaners, it’s, you know, maintenance guys. Anyone who identifies as Aboriginal is welcome to go to that forum. And the CEO turns up… you have got the CEO of [health service] sitting at the table talking to Aboriginal staff at all levels*.(Participant 8, Manager, Indigenous)

Service B had analysed their patient data to understand the Indigenous population in their catchment and referral system. They expected to find that the majority were coming from a specific region, which would help them to identify Indigenous stakeholders with whom to begin building relationships. However, this was not the case, and one staff member talked about the difficulty of building relationships with Indigenous services and organisations within their catchment, when their catchment was discovered to be nationwide. 


*When we did our first data analysis I thought, ‘Alright, we are going to find a big cohort around <regional town>’ or something like that ‘and that will be our beginning link’, but, no… the patients were scattered across the whole state and the whole country. So, the difficulty then is how do you extend your tentacles into every single area to build those relationships?*
(Participant 32, Researcher, non-Indigenous)

#### 3.2.2. Action 1.2: Addressing Health Needs of Aboriginal and Torres Strait Islander People and Action 1.4: Implementing and Monitoring Targeted Strategies

Both Services A and C made improving Indigenous health outcomes a key commitment of their strategic plans. Service A listed Indigenous Australians as one of five priority populations of patients on their Strategic Service Plan, whereas Service C’s Health Service Plan made “Closing the gap in health outcomes for Aboriginal and Torres Strait Islander People” one of five key directions for the future development of services. This commitment was felt on the ground, with multiple staff at both services stating that their service was *“highly committed to Aboriginal health”* (Participant 10, Project Officer, non-Indigenous) and observing *“they put in a lot of time and effort with their policies, like, ‘you include Aboriginal health when you are doing something’”* (Participant 4, ILO, Indigenous). 

*We certainly do have the support of executive members to do what we need to do to have good outcomes <for Indigenous patients>. That doesn’t necessarily translate to a bottomless bucket of money, but, you know, we have opportunity to within the current structures and the framework is that we can be bold to do more*.(Participant 7, Manager, Indigenous)

Both services had an active Reconciliation Action Plan (RAP) and staff were aware of its existence. Progress on the RAP was on the agenda and discussed at every Executive meeting. Service A is operated by a national not-for-profit healthcare organisation, and the organisation has a long-standing commitment to their RAP, with their first RAP implemented over a decade ago. Their Stretch RAP was described as *“the overarching strategy document to empower Australia’s First Peoples”* for all health care facilities that the national organisation operates. In 2015, Service A published an “Aboriginal and Torres Strait Islander Employment Plan”. Connected to the RAP, this plan outlined objectives and an implementation framework for the whole organisation. Each section contains multiple action items with assigned responsibilities, timelines and targets. Service B does not have a RAP; furthermore, strategic plans made minimal mention of Indigenous health. 

#### 3.2.3. Action 1.21: Improving Cultural Competency

Both Services A and C had made a consistent and ongoing effort to develop and support a strong Indigenous health workforce [21]. In contrast, staff at Service B described recruiting and retaining of Indigenous health staff as a challenge for their health service, with no Indigenous employment strategy and minimal cultural awareness training. See this article’s section on Priority 5a of the Cancer Framework for more information. 

#### 3.2.4. Action 1.33: Creating a Welcoming Environment

Service A and Service C had made an effort to create an environment where Indigenous people felt safe, comfortable and accepted. Physical representations of respect included Acknowledgement of Country plaques displayed at building entrances, Aboriginal and Torres Strait Islander flags and local artwork prominently displayed throughout the hospital and Indigenous health promotion posters featuring photos of Indigenous staff members. One family member observed *“It does feel welcoming. You know that people are recognising it, acknowledging it.”* (Carer 2 talking about the Indigenous flags and artwork at Service C).

In contrast, several staff at Service B observed that more could be done to create a welcoming environment. Staff commented that a recent move to a new, purpose-built, externally owned hospital building had inadvertently created a less welcoming physical environment for Indigenous patients. Indigenous flags, Acknowledgement of Country plaques and Indigenous artwork were no longer displayed. A patient participating in a cultural safety audit described the hospital *as “very white, glary”*, with a director at the service acknowledging: *“It is very white. The inside is white. But… is it a very white culture as well? And I think that person who was interviewed meant both. And so we need to do things. Like, we don’t have flags.”* (Participant 25, Director, non-Indigenous).

All three services celebrated significant events on the cultural calendar, such as Reconciliation Week and NAIDOC week (annual Australian observances celebrating the history, culture and achievements of Indigenous peoples). A staff member at Service A reported that organising cultural events was a shared responsibility and a way of building relationships between Indigenous staff and non-Indigenous staff: *“It’s not just ALOs doing it. There’s a whole lot of staff from the hospital helping to run it. And the beauty of it is that working together experience”* (Participant 10, Project Officer, non-Indigenous). 

The role of traditional practices was considered by Service C, with oncologists talking about the importance of encouraging patients in a *“non-judgmental and most sincere way”* to tell them if they were using bush medicine, in order to prevent harmful interactions with prescription medications. An oncologist described including the bush medicine practitioner in consultations. An Elder and ILO at Service C conducted smoking ceremonies at significant events, such as when the new palliative care centre opened. 

#### 3.2.5. Action 5.8: Identifying People of Aboriginal and/or Torres Strait Islander Origin

Both Service A and Service C had made a sustained effort to correctly identify and record Indigenous status, with processes on accurately identifying and recording Indigenous status, training for staff on how to ask identification questions and data systems enhancements made to flag Indigenous status in clinical information systems. However, staff at Service B thought that there was a need to improve identification rates at their service, with staff not routinely asking whether patients identified and Indigenous status not flagged in clinical information systems. See this article’s section on Priority 7a of the Cancer Framework for more information.

## 4. Discussion

This paper explores the performance of three services when compared to two national best practice guidelines: the Cancer Framework and the User Guide. Although all three services had previously been identified as “high performing” in their care for Indigenous cancer patients and their families, they performed very differently and were observed to have different strengths and weaknesses when the data were analysed through the dual lenses of the Cancer Framework and the User Guide. This study suggests that even “high performing” tertiary services struggle with the first four priorities of the Cancer Framework (summarised as Education, Prevention, Screening and Early Diagnosis). This invites further discussion of what is realistic for tertiary services to do in the area of working more upstream. Of the three services, Service C performed the best on the Cancer Framework with a rating of green for seven priorities, one amber rating and one red. Service A performed moderately with a rating of green for four priorities, one amber rating and four red. Service B performed the least well with a rating of amber for one priority and eight red. Services A and C, which performed better on the Cancer Framework, also performed strongly against the User Guide, with a rating of green for all six actions, whereas Service B had a rating of green for only one action, and five red.

The Cancer Framework has a strong focus on upstream cancer care, with four of the seven priorities focused on activities that take place before patients commence treatment [15]. However, only Service C scored a rating of green against any of the upstream priorities. This suggests that Education, Prevention, Screening and Early Diagnosis were not a priority for the cancer services at the time of the case study. Furthermore, if even these services that are “high performing” with respect to their Indigenous cancer patients struggle to deliver in these areas, it suggests that other tertiary cancer services may also be struggling in these areas. This begs the question, should tertiary cancer services have a responsibility to deliver education and prevention programs to Indigenous communities, promote screening participation within the Indigenous population and improve the linkages with primary health care that aid early diagnosis to their Indigenous cancer patients? The treatment-focused Cancer Pathway may be a more appropriate framework for measuring the performance of tertiary cancer services in caring for their Indigenous patients.

Nevertheless, if services are to reduce inequalities in cancer mortality, they must look beyond their traditional biomedical treatment focus and target structural inequities by improving accessibility to their services and committing to a more holistic approach to Indigenous health [14,15,27,28,29,30,31]. Comments made by staff at all three services suggest that there is already considerable interest and support from staff on the ground. However, sustained efforts in these areas require leadership, new projects and additional resources to provide the driver, which may not be a priority for tertiary services already stretched to capacity. Of the three services, Service C was the most active in the upstream priorities. However, most of their activities in education, prevention and screening appeared to be driven by a small number of staff champions. Supportive policies and additional funding may provide a driver for organisations to make formal commitments towards improvements in this space [20]. Furthermore, health services do not have to tackle these areas alone. Partnerships between cancer services and organisations who are already working in this space, such as Cancer Councils, Indigenous health organisations and primary health care providers, have the potential to strengthen the delivery, reach and cultural safety of education, prevention and screening programs and to improve the linkages that aid early diagnosis [30,32].

The barriers to accessing specialist and hospital care reported by Indigenous participants in this study indicate that there is a need to tackle structural inequality at the national level as well as at the health service level. Recent articles conclude that inequitable cancer outcomes are caused in part by the accumulated disadvantage that Indigenous Australians face across multiple areas including health, housing, education and justice [31,33]. Policies targeting structural and social inequities (such as Closing the Gap) are likely to increase health care utilisation among Indigenous Australians [33]. Closing the Gap is an Australian government strategy adopted in 2008 that aims to improve the following outcome areas: health and wellbeing, education, employment, justice, safety, housing, land and waters, languages and digital inclusion [34].

It is not surprising that services which performed better on the Cancer Framework also performed strongly on the User Guide. Cancer services and oncology wards do not exist in isolation; they are directed and influenced by the leadership and culture of the health services they are part of. Unlike the cancer-specific Cancer Framework, the User Guide aims to improve the quality of care and health outcomes for Indigenous people at the health service level. We suggest that Services A and C had the ability and support to implement Cancer Framework priorities because the health services had made a long-term and ongoing commitment to improving Indigenous health: they were working in partnership with Indigenous communities (Action 2.13 in the User Guide) and had made Indigenous health a priority, which was implemented and monitored through policies and procedures (Actions 1.2 and 1.4 in the User Guide) [18]. Furthermore, due to the overlap between the User Guide and the Cancer Framework (see Figure 1), having systems, processes and training in place at the organisational level to improve the cultural competency of the health service (Action 1.21) and to accurately identify Indigenous staff (Action 5.8) aided the cancer service in meeting Cancer Framework priorities 5a (ensure a skilled and caring workforce with effective cross-cultural communication skills) and 7a (enhance data systems to inform better outcomes) [15,18].

In contrast, the performance of Service B against both sets of guidelines shows how challenging it can be to sustain “high performance” when trying to provide culturally safe care for Indigenous people. Although identified as “high performing” during the initial survey and interview, with a number of staff championing Indigenous health and a number of projects underway, by the time the case study took place Service B had undergone a number of organisational changes, including relocating to a new building, organisational restructure and the turnover of a number of Indigenous staff and non-Indigenous staff who championed Indigenous health. We suggest that because Service B did not have Indigenous-focused policies and processes in place (as specified in Actions 1.2 and 1.4 of the User Guide), combined with an inability to recruit or retain Indigenous staff (Action 1.21) and difficulty identifying Indigenous patients (Action 5.8), the loss of key staff who championed Indigenous health meant that projects and initiatives that were underway stopped or slowed. While we observed tremendous goodwill from staff during the case study, there appeared to be a lack of leadership and guidance on how to direct it. Service B is not alone in this, with many busy services wishing to perform better, but struggling to make the investments of time, funding and resources required to bring meaningful change to Indigenous health [19]. For services to make real progress, strong leadership is required by the health service executive and the commitment to Indigenous health needs to be prioritised in the service’s values or mission statement, as was observed with Services A and C and has been reported previously [35,36,37].

### Limitations

One of the original intentions of this study was to explore connections between cancer services and primary health care, which may have provided additional information on Cancer Framework Priority 4 (Early Diagnosis). However, we experienced delays with this project which reduced our capacity and timeframes, and consequently this area was not able to be explored.

While we made every effort to gather an accurate and complete picture of the cancer services by interviewing both Indigenous and non-Indigenous staff from a diverse range of professions, including upper management, clinicians and support staff, services may have initiatives that were not observed or reported during the case studies. Furthermore, services are dynamic with respect to initiatives and programs, while data collection for the case studies occurred over a discrete period. We are aware of new initiatives or changes at all three services that have been implemented since the case studies with the aim of improving care and support for Indigenous patients and their families. Since the case study, Service B appears to have made a renewed commitment to improving Indigenous health outcomes as outlined in Appendix A.

We were only able to interview eight Indigenous people affected by cancer because participants were only approached if it was felt their mental health and physical well-being would not be compromised by participating in this study. Several Indigenous patients were unavailable on the day of their scheduled interview due to poor health or discharge from hospital.

## 5. Conclusions

Health services and cancer services wishing to improve patient outcomes and experiences of care for Indigenous people affected by cancer have much to learn from the three services in this article. This article highlights the importance of a whole-of-organisation approach when addressing and embedding the six actions of the User Guide. Service B shows how easily organisational change can disrupt initiatives to provide culturally safe care to Indigenous patients. Goodwill alone or a small number of staff champions cannot make a service culturally safe. Long-term, sustainable change requires an understanding of Aboriginal culture, as well as a willingness to learn, partnerships with Indigenous people and communities, the prioritisation of Indigenous health, the involvement of Indigenous staff in implementation efforts and ongoing monitoring of strategies—approaches laid out in the User Guide and demonstrated by Services A and C. Once a health service has implemented the User Guide, it already has some of the culture and support in place required to implement the Cancer Framework or Cancer Pathway.

This article is significant because it is the first to compare the performance of tertiary cancer services against both the Cancer Framework and the User Guide, two key policy documents. While two services performed well against the User Guide, all three struggled with the upstream elements of the Cancer Framework, suggesting that the treatment-focused Cancer Pathway may be a more appropriate framework for tertiary services. While tertiary cancer services could play an important role in delivering education and promoting prevention and screening participation which could in turn improve their linkages with primary health care providers to aid early diagnosis, it is evident that much more discussion is needed. If additional funding was made available for these upstream functions, would it be more effectively spent supporting primary or tertiary health care providers or community health organisations to work in this space? Partnership approaches may offer one way forward, with tertiary health services strengthening relationships with relevant organisations such as Cancer Councils, Indigenous health organisations and primary health care providers to enhance the delivery, reach and cultural safety of their services. Finally, additional Indigenous-led research is needed to identify and evaluate successful cancer service delivery programs for Indigenous Australians. This will support health services to continue improving patient outcomes and experience of care for Indigenous cancer patients and their families.

## Data Availability

The datasets generated and/or analysed during the current study are not publicly available due to small participant numbers and protection of confidentiality. Aggregate data are available from the corresponding author on reasonable request.

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
