# Peer review of "Putting Policy into Practice: How Three Cancer Services Perform against Indigenous Health and Cancer Frameworks"

_ijerph, 2022, doi:10.3390/ijerph19020633_

Round 1

Reviewer 1 Report

This paper is very well written and I appreciate their work. Conclusions are supported by data, however should be reduced to essential. 

Author Response

We thank the reviewer for these comments.

We have gone through the manuscript and attempted to shorten/summarise the content where possible. However, while tightening up the manuscript, in responding to the Academic Editor’s comments and the detail requested by the Reviewers’ comments, the overall length of the paper has not been reduced.

Reviewer 2 Report

It is an interesting investigation study

It is not clear what kind of questionnaire was used for the participants' interview.

The work needs to clarify the purpose of the investigation, 
was the purpose also to identify people at risk of cancer and to set up screening programmes? 

Author Response

We thank the reviewer for their comments. 

Question 1: 
As stated on lines 163 and 170: “In-depth interviews were conducted…” these were “Semi-structured interviews” (line 176) and “conducted with the use of the appropriate interview guide” (line 182).  
Also on lines 192-197: “Indigenous people affected by cancer were asked about their experiences at the cancer centre, including issues experienced, and any suggestions for changes or improvements. Hospital staff were asked a range of questions regarding existing policies and programs to improve engagement with Indigenous people, with particular emphasis on cancer patients, including questions on primary prevention and education programs, cultural awareness programs, cultural identifiers and Indigenous staff employment strategies.”

However, we have added the following about the interview guides to make this clearer: 
The research team created two interview guides (one for hospital staff and one for Indigenous people affected by cancer) which guided the direction of the interviews; however additional questions were occasionally added in response to individual roles and answers to questions.

Question 2: 

No that was not the purpose of the investigation. 

We believe the purpose of the investigation was stated clearly in lines 105-106: 
“This study forms part of a national investigation to identify and describe cancer services providing treatment to Indigenous cancer patients in Australia.”

And also in lines 93-101: 
“The aim of this paper was to explore how three services, which had previously been identified through a comprehensive process as “high performing” in the care they provide to their Indigenous patients and family, measured up when compared to two national best practice guidelines: the Cancer Framework and the User Guide. We wished to identify the areas where the three services are doing well and highlight what activities they are performing in those areas, as this may benefit other services wishing to improve their performance on one or both guidelines. However, we also wished to explore whether there are sections in the guidelines where even high performing services struggle, and to identify where additional support or attention may be required.”

Reviewer 3 Report

The research used method of intrinsic case study to analyze the selected three Australian cancer services for indigenous people, which including in-depth interviews conducted with relevant hospital staff (n=35, 10 participants were employed in Service A, 11 participants worked in Service B, and 14 participants worked at Service C) as well as in-depth interviews that conducted with indigenous patients with cancer (n=5; four male) and affected family members (n=3; all female). All the interviews were analyzed and scored with a traffic light system according to the 7 priorities of the Cancer Framework and the 6 actions of the National Safety and Quality Health Service Standards (NSQHS) User Guide. The authors concluded that “While two services performed well against the User Guide, all three struggled with the upstream elements of the Cancer Framework, suggesting that the treatment-focused Optimal Care Pathway for Aboriginal and Torres Strait Islander People with Cancer (Cancer Pathway) may be a more appropriate framework for tertiary services. This article highlights the importance of a whole-of-organisation approach when addressing and embedding the six actions of the User Guide. Health services which have successfully implemented the User Guide are in a stronger position to implement the Cancer Framework and Cancer Pathway.”

This is a well-writing manuscript following the 2 previous articles: International Journal of Environmental Research and Public Health 2018, 15, 717 (doi:10.3390/ijerph15040717) and PLOS ONE 2020, 15, e0239207 (doi:10.1371/journal.pone.0239207). However, I hope they described more procedures in detail. For example, line 105, “Surveys were completed for 58 of the 125 public cancer treatment centres.” How were the 58 cancer treatment centers selected? Randomly? Or by some rules?

In addition, similarly, why were the 35 staffs selected? Because there were thousands of staffs. Only 5 patients and 3 affected family members participated in-depth interviews. Did the participants be represented enough the study goal? I think authors may discuss this point or presented in limitation.

Minor:

Line 18, abbreviate NSQHS appeared firstly should include full term. I suggested better to use “National Safety and Quality Health Service Standards (NSQHS) User Guide”.

Author Response

We thank the reviewer for their comments. 

Question 1: 

We contacted all 125 public cancer treatment centres and asked them to complete the survey, but only 58 centres completed the survey. Our procedure is described in more detail in our 2018 article published in Aust N Z J Public Health (https://doi.org/10.1111/1753-6405.12843) and referenced in the preceding sentence on line 106. However, to make this clearer we have changed the sentence to read: 
Surveys were completed by 58 of the 125 public cancer treatment centres contacted.

Question 2: 
As stated in the article at lines 147-150: “Hospital staff (Indigenous and non-Indigenous) were eligible to participate if they cared for or supported Indigenous cancer patients or if they held a leadership role in the care of Indigenous patients at one of the three sites in the case study.” 
Also from line 154: “Recruitment was purposive, with staff, Indigenous cancer patients and family members identified and recruited in-person by local health service staff within each participating service. The participants were generally selected in conjunction with the site investigator”

Participant recruitment was covered in considerable depth in our 2021 article published in BMC Health Services Research (https://doi.org/10.1186/s12913-021-06535-9). 

This limitation was covered in our 2021 article published in BMC Health Services Research (https://doi.org/10.1186/s12913-021-06535-9). 
However, I have added the following to the current article in the Limitations section: 
We were only able to interview eight Indigenous people affected by cancer because participants were only approached if it was felt their mental health and physical well-being would not be compromised by participating in this study. Several Indigenous patients were unavailable on the day of their scheduled interview due to poor health or discharge from hospital.

Minor: 

Done.